# Nutritional Support in Malnourished Outpatients with Chronic Obstructive Pulmonary Disease (COPD): A Randomized Controlled Pilot Study

**DOI:** 10.3390/nu16111696

**Published:** 2024-05-30

**Authors:** Valerie Conway, Craig Hukins, Stacey Sharp, Peter F. Collins

**Affiliations:** 1Nutrition and Dietetics, Brisbane South Chronic Disease Service, Metro South Hospital and Health Service, Wynnum, QLD 4178, Australia; valerie.conway2@health.qld.gov.au; 2Department of Respiratory and Sleep Medicine, Princess Alexandra Hospital, Brisbane, QLD 4102, Australia; craig.hukins@health.qld.gov.au; 3Department of Nutrition and Dietetics, Logan Hospital, Meadowbrook, QLD 4131, Australia; stacey.sharp@health.qld.gov.au; 4Nutrition and Dietetics, Sydney Nursing School, Faculty of Medicine and Health, The University of Sydney, Sydney, NSW 2006, Australia; 5Charles Perkins Centre, The University of Sydney, Sydney, NSW 2006, Australia

**Keywords:** malnutrition, COPD, nutritional support, chronic obstructive pulmonary disease, oral nutritional supplements

## Abstract

(1) Background: The evidence for nutritional support in COPD is almost entirely based on ready-to-drink oral nutritional supplements (ONSs). This study aimed to explore the effectiveness of powdered ONSs alongside individualized dietary counseling in the management of malnutrition. (2) Methods: Malnourished outpatients with COPD were randomized to receive either routine care (Group A: counseling + recommended to purchase powdered ONSs) or an enhanced intervention (Group B: counseling + provision of powdered ONSs at no cost to the patient) for 12 weeks. Outcomes of interest were nutritional intake, weight status, and quality of life. (3) Results: A total of 33 outpatients were included, categorized as follows: Group A (*n* = 21); Group B (*n* = 12); severely malnourished (*n* = 9), moderately malnourished (*n* = 24), mean BMI 18.0 SD 2.5 kg/m^2^. No differences were observed between groups at baseline or at week 12; however, analysis of the whole cohort (Group A + B) revealed nutrition intervention resulted in significant improvements in protein intake (+25.4 SD 53.4 g/d; *p* = 0.040), weight (+1.1 SD 2.6 kg; *p* = 0.032) and quality of life (−4.4 SD 10.0; *p* = 0.040). Only 41.2% of Group A and 58.3% of Group B reported consuming ONSs at week 12. Adherence to ONSs was associated with weight gain (+1.9 SD 2.5 kg vs. +0.4 SD 2.5 kg; *p* = 0.098). (4) Conclusions: Nutritional support results in significant improvements in nutrition status and quality of life in malnourished outpatients with COPD. However, improvements are associated with adherence to ONSs, suggesting the type of ONSs and how they are provided are important considerations in clinical practice and future studies.

## 1. Introduction

The nutritional and metabolic implications of COPD have long been the subject of extensive scientific research due to the progressive irreversible nature of the disease [1]. Unintentional weight loss is common in patients with COPD, and up to 40% have been found to experience clinically relevant losses [2], which have also been associated with exacerbation frequency [3]. Disease-related malnutrition is a common problem in COPD, with studies using the Subjective Global Assessment (SGA) diagnosing in excess of 50% of patients as malnourished [4,5]. Malnutrition has also been found to be independently associated with significantly increased healthcare utilization, healthcare costs, and mortality [6]. However, its etiology is incredibly complex and multi-factorial, including an inability to consistently meet nutritional requirements due to the symptomology related to the disease (e.g., shortness of breath, anorexia, early satiety) [7], which are further compounded during periods of hospitalization [8]. Recently, factors such as social deprivation have also been found to be associated with malnutrition risk and healthcare use [9,10] and are likely to be an important consideration in formulating effective nutritional intervention strategies. 

Systematic reviews and meta-analyses have demonstrated that if malnutrition or malnutrition risk are identified in outpatients with stable COPD (non-exacerbating), they are amenable to nutritional treatment, resulting in significant improvements in nutritional intake and status [11], as well as improvements in functional capacity and quality of life [12]. However, the current evidence base is almost entirely based on ready-to-drink oral nutritional supplements (ONSs) and is lacking other first-line nutritional intervention strategies commonly used in routine clinical practice (e.g., food fortification and powdered ONSs), either alone or in combination (multi-modal). However, one study included in the reviews did involve dietary counseling by a dietitian and the provision of a 6-month supply of milk powder, which resulted in significant improvements in body weight and quality of life [13]. More recently, a randomized trial involving a 12-month intervention of ready-to-drink ONSs or high-energy, high-protein snacks found both interventions associated with improvements in weight and quality of life [14]. However, while ONSs are often subsidized within many health care systems, this is often not the case for milk powder or food items. Whether implementation of these approaches allowing the provision and prescription of food items in the community is feasible and acceptable remains to be established. There also remains very limited evidence for the use of powdered ONS and food fortification in the management of malnutrition in COPD, but with increased financial pressures, this has seen healthcare systems make stronger recommendations for the use of this approach as a first-line intervention over ready-to-drink ONSs [15]. Due to a lack of research exploring the effectiveness of current first-line dietetic practices in the treatment of malnutrition in COPD, this study aimed to explore the effectiveness of a multi-modal nutritional intervention commonly used as a first-line treatment for malnutrition (powdered ONSs and individualized dietary counseling provided by a dietitian).

## 2. Materials and Methods

### 2.1. Recruitment and Randomisation

Patients attending respiratory outpatient clinics at the Princess Alexandra Hospital, Brisbane, between April 2015 and June 2017 with a confirmed diagnosis of COPD (post-bronchodilator FEV_1_/FVC < 0.7) and identified as at risk of malnutrition using the Malnutrition Screening Tool [16] and/or had a BMI ≤ 21 kg/m^2^ [17] were referred to Brisbane South Chronic Disease Service, Metro South Hospital and Health Service. Nutrition assessment and diagnosis of malnutrition were completed by a dietitian using the Subjective Global Assessment (SGA) [18]. Outpatients diagnosed with mild/moderate (category B) or severe malnutrition (category C) who met the inclusion criteria, were invited to participate in the study. Patients unable to provide informed consent, who were already receiving nutritional support, or who had an underlying condition that would likely impact future nutritional status and/or response to nutritional support were excluded from the study (e.g., malignancy, congestive cardiac failure). In addition, patients had to have been free from an exacerbation of COPD for at least 4 weeks prior to enrollment (stable COPD), and informed written consent was obtained from all patients prior to randomization. 

### 2.2. Intervention

Outpatients were randomized to either Group A (routine care) or Group B (ONSs). The intervention provided to Group A represented routine care currently offered by the dietetics service and involved an experienced clinical dietitian providing individualized dietary counseling, provision of a standard written nutrition educational resource around high-energy, high-protein nourishing snacks, food fortification, and a recommendation to purchase a powdered ONS, Sustagen^®^ Hospital Formula (Nestle), available from commercial pharmacies. Patients were advised to consume two servings (200 mL per serving) of the powdered ONS per day during the intervention period, preferably between meals. The ONS, when made up with water, provided a total of 450 kcal and 27.6 g protein per day, but patients were recommended to prepare the ONS with 200 mL of full-fat milk where possible (400 mL milk: +264 kcal, +13 g protein). If the prescribed servings were consumed, this would result in an additional 714 kcal and 40 g protein per day. Average cost per commercially purchased tin would have been $28 AUD, but participants in Group A were able to access the supplement at a subsidized rate of $10 AUD through Queensland Health. The decision to use Sustagen^®^ was based on it being a powdered ONS product that is frequently used and recommended as part of local practice, and, therefore, patients were likely to be familiar with it. Outpatients randomized to Group B (ONS) received the same routine care provided to Group A with the exception that one tin (840 g) of the ONS was provided to the patient per week at no cost, enough to make two serves of Sustagen^®^ per day. Outpatients in Group B were offered a choice of flavors based on taste preference (chocolate, vanilla, or neutral). The intervention was completed over 12 weeks, involving a baseline assessment with a dietitian and follow-up face-to-face visits at weeks 6 and 12 between 30 and 60 min in duration. Individualized dietary counseling was provided at baseline and reinforced, and tailored as needed, at week 6. Given the high burden associated with COPD and malnutrition, where outpatients were unable or unwilling to physically attend the outpatient nutrition and dietetics clinic, home visits were completed by the research dietitian.

### 2.3. Outcome Measures

All data were collected by the research dietitian during assessment visits conducted either at the outpatient clinic or the patient’s home (baseline, week 6, and week 12). Nutritional intake data were collected at each time point using a structured 24 h dietary recall method, and adherence to ONS was collected at follow-up visits. Dietary intake data were entered and analyzed independently, including blinding to the intervention groups using Foodworks (version 8.0 Xyris). Weight was measured at each visit without shoes and in light clothing using a scale Model 876 (Seca, Hamburg, Germany) to the nearest 0.1 kg. Height was measured using a portable stadiometer, Model Seka 213, to the nearest cm. Handgrip strength was assessed using a hydraulic hand dynamometer, Model SH5001 (Saehan, Busan, Republic of Korea), with the outpatient in a seated position (resting on arm of chair). The St. George’s Respiratory Questionnaire (SGRQ) [19] was used to assess the patient’s health-related quality of life. Data was also recorded on COPD disease severity [20], nutritional status (BMI and SGA), whether the patient lived alone, and if they received support from a caregiver. 

Ethical approval was granted by the Princess Alexandra Hospital Human Research and Ethics Committee, Brisbane Australia (HREC/14/QPAH/274) ANZ trial reference number: ACTRN12619001586101.

### 2.4. Sample Size Calculation and Statistical Analysis

In malnourished patients, a weight gain of 2 kg has been demonstrated to be feasible with predominantly ready-to-drink ONSs [11], and this level of weight gain was associated with improvements in functional capacity and quality of life [12]. In order to achieve the statistical power to detect differences in primary endpoints (quality of life and body weight), a target sample size of 100 patients was identified. This was calculated, aiming for 80% power, significance set at *p* < 0.05, and a difference in total SGRQ score of 4 (reduction in total score ≤ 4 = minimally important clinical difference). All patients who completed the week 6 visit were included in the analysis, with full analysis undertaken on those patients who completed all 3 visits. Data were analyzed, once all patients had completed the study, using SPSS statistical software package version 23.0 (Chicago, IL, USA). Continuous variables, such as nutritional intake, body weight, quality of life and corresponding changes, are presented as mean ± standard deviation (SD), unless otherwise stated. Categorical variables, such as malnutrition category and COPD disease severity, are presented as *n* (%). Differences between categorical and continuous variables were evaluated by comparing their mean ± SD using a one-way ANOVA test. Changes in outcomes of interest within intervention groups were assessed using paired-samples *t*-tests, and pooled analyses of changes (Group A + B) were conducted using one-sample *t*-tests. A *p*-value of ≤0.05 was considered statistically significant, and changes in body weight (≥2 kg) and quality of life (SGRQ ≤ 4) were considered clinically relevant. Data is presented in accordance with the CONSORT statement [21] and with careful attention to transparency and clarity when it comes to the reporting of data relating to oral nutritional supplements (Template for Intervention Description and Replication (TIDieR)) [22].

## 3. Results

### 3.1. Patient Characteristics

Thirty-three outpatients with confirmed COPD and a diagnosis of malnutrition were randomized (Figure 1). Twenty-one patients received routine nutrition and dietetic care (Group A), and 12 patients received routine dietetic care plus the provision of ONS (Group B). At baseline, nutritional assessment diagnosed 24 outpatients as moderately malnourished (SGA B) and 9 as severely malnourished (SGA C), with no differences observed between the two groups (Table 1). The mean BMI for the cohort was 18.0 SD 2.5 kg/m^2^ (range 13.1 to 25.0 kg/m^2^), and most of the patients were classified as having severe or very severe COPD (moderate 15.2%, severe 29.4%, and very severe 45.5%). One-third of the outpatients were receiving support from a caregiver (30.3%), and, in terms of income, patients were either reliant on a government (54.5%) or disability support pension (45.5%). Most outpatients in the study required a home visit by the dietitian due to being unwilling or unable to attend the outpatient clinic (72.7%). Outpatients requiring assessment visits to be conducted at home did tend to be older and with greater nutritional depletion (BMI 17.6 SD 2.6 kg/m^2^ versus 19.1 SD 1.9 kg/m^2^; *p* = 0.132) and a higher prevalence of severe malnutrition, with 88.9% of patients classified as SGA-C visited at home. Otherwise, there were no differences in the characteristics between the two groups at baseline (Table 1), and this remained the case at the end of the 12 week nutritional intervention. 

Analysis of the whole cohort (Group A + Group B (n 24 complete data), paired *t*-test) revealed nutritional support was associated with an improvement in body weight above baseline at week 6 (+0.5 SD 1.7 kg, *p* = 0.079), and this reached statistical significance after 12 weeks of nutritional support (+1.1 SD 2.6 kg, *p* = 0.032). This was accompanied by a significant and clinically meaningful improvement in patient-reported quality of life (St. George’s Respiratory Questionnaire score −4.4 SD 10.0; *p* = 0.040, with a reduction in score ≥ 4 indicating efficacy). 

### 3.2. Energy and Protein Intake

Pooled analysis revealed a large increase in energy intake from baseline both at week 6 (+299 SD 917 kcal/d, *p* = 0.072) and at week 12 (+399 SD 1228 kcal/d, *p* = 0.091), but this did not reach statistical significance due to considerable variation around the mean change observed in food intake and the small sample size. At week 6, the increase in energy could be entirely attributed to the ONSs (100% additional energy), and at the end of the intervention, 34% of the additional energy appeared to be consumed through food and 66% consumed via the ONSs (Figure 2). Analysis of nutritional intake in the whole cohort and adjusting for only those reporting to be consuming ONS, revealed no significant impact on energy consumed via food at either time point. 

Protein intakes were found to be significantly improved above baseline both at week 6 and week 12 (Figure 3), with 67% of the additional protein attributable to the powdered ONSs. When analyzing the data, including only those outpatients reporting to be consuming ONSs at week 12 (n = 14), the mean improvement in daily protein intake was +31.0 SD 19.8 g/d, 95% CI +19.6 to 42.5 g/d, *p* < 0.001. The mean protein intake at week 12 for the cohort was 1.9 SD 1.0 g protein/kg body weight/day.

### 3.3. Body Weight

At the end of the nutritional intervention, both groups had increased body weight above baseline, with no differences between the groups (Group A: +0.95 SD 2.2 kg versus Group B: +1.3 SD 3.2 kg; *p* = 0.203). Overall, the cohort gained a significant amount of weight, +1.1 95% CI +0.1 to +2.1; *p* = 0.032. 

### 3.4. Adherence to Oral Nutritional Support (Dietary Advice and ONSs)

There were no differences in the baseline characteristics between those outpatients reporting as consuming ONSs and those who were not (age, gender, BMI, nutritional intake (energy and protein), and COPD disease severity). Only half of outpatients reported to be consuming the ONSs at week 6 (51.5%) and at week 12 (48.3%) of the intervention, and there were notable differences between the two groups. Those outpatients who were provided the ONSs for the purpose of the study reported a higher consumption at week 6 than those who were recommended to purchase the same ONSs themselves (Group A: 42.9% versus Group B: 66.7%, Chi-square *p* =0.188). This self-reported adherence trend persisted until the end of the 12-week intervention (Group A: 41.2% versus Group B: 58.3%, chi-square *p* = 0.362), with the overall adherence to ONSs at week 12 being less than half (48.3%). Although adherence to ONSs and weight change were not significantly different between the two intervention groups, self-reported adherence to ONSs (supplied or purchased) was associated with a greater improvement in body weight at the end of the intervention (+1.9 SD 2.5 kg versus −0.3 SD 2.5 kg; *p* = 0.085). Although the difference in weight change between the two groups was not statistically significant, this was due to the sample size, and such an improvement has been found to be of clinical relevance [12].

Outpatients who reported to be consuming ONSs at the end of week 12 did tend to be older and have a slightly lower nutritional intake at baseline, but the differences were not significant. Other noteworthy trends were that patients receiving support from a caregiver were more likely to be consuming ONSs at week 12 than those not in contact with a caregiver (62.5% versus 42.9%, X^2^ *p*-trend = 0.427), and reliance on ONSs appeared to increase with increasing COPD disease severity (moderate: 25.0%, severe: 45.5%, very severe: 57.1%; X^2^ *p*-trend = 0.511). When analysis was limited only to those patients reporting to be consuming ONSs at week 12 (n = 14), there was a small reduction in energy intake from food (−108 SD 821 kcal/d, 95% CI −582 to +367 kcal/d, *p* = 0.632), but with a significant amount of energy provided by ONS (+543 SD 353 kcal/d, 95% CI +339 to +746 kcal/d, *p* < 0.001). 

### 3.5. Handgrip Strength and Quality of Life

Handgrip strength measured in both hands decreased in both groups from baseline to week 12. The decline was larger in the right hand (90.9% right-hand dominant) but was not significantly different between the two groups (Group A −5.8 SD 15.0 versus −4.7 SD 8.5 kg; *p* = 0.826). Combined analysis (Groups A + B) of change in handgrip strength from baseline to week 12 revealed an overall decline in the left handgrip (−2.0 SD 10.3 kg; *p* = 0.304) and a significant decline in the right handgrip (−5.3 SD 12.5 kg; *p* = 0.030). There was no association between handgrip strength and other outcomes of interest. 

Quality of life was significantly improved for the cohort (Group A and Group B: SGRQ total score −4.4 95% CI −0.2 to −8.7; *p* = 0.040), and this crossed the threshold (≤4) for minimally important clinical difference. The magnitude of the improvement in quality of life observed increased with malnutrition severity, with the greatest improvement seen in those with severe malnutrition (SGA A: −1.4 SD 3.9, SGA B: −4.2 SD 9.4, SGA C: −8.1 SD 15.7; *p* = 0.587). 

### 3.6. Change in Malnutrition Diagnosis

At baseline, 72.7% of the cohort were diagnosed with moderate malnutrition and 27.3% with severe malnutrition, according to the SGA. However, at the end of the intervention, 20.7% of patients were classified as having a malnutrition diagnosis resolution (SGA-A), 62.1% had moderate malnutrition, and 17.2% had severe malnutrition. There was a non-significant trend for the severely malnourished patients to gain more weight. Using the pre-defined cut-off of ≥2.0 kg weight change to identify responders to nutritional treatment, responders tended to be older (72.7 SD 8.8 years versus 62.1 SD 13.3 years, *p* = 0.105), with more significant nutritional depletion at baseline (BMI 17.1 SD 1.9 kg/m^2^ versus 19.3 SD 3.6 kg/m^2^, *p* = 0.181). However, they did consume considerably more energy per kilogram body mass than those outpatients who did not gain the target weight (55 SD 20 kcal/kg body weight/day versus 46 SD 18 kcal/kg body weight/day, *p* = 0.261) and increased their energy intake above baseline by more than double (+626 SD 1164 kcal/kg body weight/day versus +260 SD 1278 kcal/kg body weight/day, *p* = 0.446). Comparison of energy intakes in relation to body weight according to changes in malnutrition diagnosis revealed a trend for a higher energy intake in those with improved malnutrition status (Figure 4). With the aim of nutritional support being to treat malnutrition, analysis of outpatients whose SGA classification remained unchanged or worsened versus those patients whose malnutrition diagnosis improved or resolved revealed a significant difference in energy intake (58 SD 16 kcal/kg body weight/day versus 44 SD 19 kcal/kg body weight/day, *p* = 0.038).

## 4. Discussion

This study found that nutritional support involving a combination of powdered ONSs and individualized dietary counseling provided by a dietitian resulted in significant improvements in nutritional intake, body weight, and quality of life. Although the weight gain observed in the current study was less than the mean weight gain reported in previous meta-analyses [11,12], it was still associated with significant improvements in self-reported quality of life beyond the minimally important difference assessed using the St. George’s Respiratory Questionnaire. Previously, this degree of weight gain in outpatients at risk of malnutrition has been found to be associated not only with improvements in quality of life but also inspiratory and expiratory respiratory muscle strength [12]. Improvements in respiratory muscle strength are likely to have a positive influence on quality of life, and, in a disease characterized by excess sputum production, strength improvements could assist patients to expectorate, helping to reduce mucus accumulation and the risk of recurrent exacerbations [23]. Although the current study did not find nutritional support to result in any improvements in handgrip strength, quality of life was significantly improved for the cohort (SGRQ total score: −4.4 95% CI −0.2 to −8.7; *p* = 0.040), crossing the threshold for minimally important clinical difference (≤4). Notably, the magnitude of the improvement in quality of life observed increased with malnutrition severity, with the greatest improvement seen in those with severe malnutrition (SGA A: −1.4 SD 3.9, SGA B: −4.2 SD 9.4, SGA C: −8.1 SD 15.7; *p* = 0.587). This should be confirmed in future studies involving a larger sample. 

Nutritional support in the form of ONSs for malnourished hospitalized patients with COPD in the United States has been found to be associated with improved body weight, increased handgrip strength, and markedly reduced mortality [24]. Although the researchers did not observe any reduction in hospital readmission rates in this study, an earlier, larger retrospective study of the use of ONSs in 14,326 hospitalized patients with COPD did find a significant reduction in length of hospital stay, the probability of 30-day readmission, and a reduction in healthcare costs [25]. Given that malnutrition in COPD presents a significant economic and operational burden to healthcare systems, associated with increased emergency hospitalization, prolonged duration of hospital stay, and early readmission [6], there is a need for future adequately powered prospective studies investigating the impact of nutritional support on health economic endpoints. To date, studies have tended to focus on stable outpatients with the disease, with limited evidence for nutritional support during acute periods of the disease beyond improving nutrition intake, which might assist with attenuating nutritional depletion [26]. 

A particular strength of the current study is that it explores the effectiveness of interventions that are reflective of routine clinical practice, and both groups received individualized dietary counseling provided by the same dietitian. The only difference between the two intervention groups was whether the powdered ONSs were provided to the patient at no cost or whether participants were advised to purchase them at a subsidized rate (reflective of local practice). The rationale for both groups being provided with individualized counseling is based on publications involving UK [13] and Vietnamese [27] cohorts of outpatients with COPD, where written educational material on food fortification, high-energy, and high-protein foods was provided to control groups but was not discussed or tailored to individuals. In the Vietnamese cohort of outpatients, no significant changes were observed in dietary intake or body weight; however, in the UK cohort, control group outpatients went on to lose weight. Given the well-known negative consequences of malnutrition in this patient group, it would not be considered clinically ethical to not provide and tailor advice for individuals identified as malnourished. In addition, this study is the first to explicitly set out to explore the efficacy of commercially available powdered ONSs in the treatment of malnutrition in COPD; therefore, providing individualized dietary counseling to both groups provided a baseline to do this. Although the Vietnamese study set out to investigate the effectiveness of tailored dietary counseling in treating malnutrition, the inclusion of patients reporting to be consuming ONSs at baseline resulted in the researchers having to stratify the analysis to establish the independent effect of dietary counseling with and without ONSs [27]. Fortunately, this was not an issue due to the large number of participants recruited to the study (n = 120). Paired analysis is not reported for baseline to 12 weeks (intervention end) for those consuming ONSs, which does make interpretation problematic. Across both intervention and control groups, consumption of ONSs appeared to add little additional value above individualized dietary counseling, although the type of ONSs consumed and the amount are not reported.

In clinical practice, dietitians typically estimate nutritional requirements using predictive equations or a ratio method (kcal/kg/day). Generally, for weight maintenance in patients with COPD, this is approximately 30 kcal/kg/day [28] and is considerably higher for those patients where weight gain is the target (45 kcal/kg/day) [29]. In the study by Nguyen et al. [27], the energy intake in the intervention group at baseline was 28.6 SD 8.2 kcal/kg/day and increased to 44.0 SD 18.6 kcal/kg/day (*p* < 0.001). This corresponded with an increase in weight from 46.0 SD 7.9 kg to 47.3 SD 8.5 kg, *p* < 0.001, and successful treatment of malnutrition, according to the Subjective Global Assessment (SGA-A), in nine (16.7%) outpatients. Similar findings were found in the current study, where 20.7% of patients were able to achieve a malnutrition diagnosis resolution. However, in comparison to those outpatients whose malnutrition diagnosis classification, according to SGA, remained unchanged or became worse, those who were no longer malnourished had a significantly higher energy intake (58 SD 16 kcal/kg/day versus 44 SD 19 kcal/kg/day, *p* = 0.0038). These findings suggest a minimum energy intake target of 45 kcal/kg/day in malnourished patients with COPD to *treat* malnutrition. It is hoped that future published nutrition intervention studies in patients with COPD will not only report on the change in energy and protein intake from baseline but also express these changes according to body weight. This is not only relevant to clinical practice but can also help inform future clinical guidelines. 

### 4.1. Limitations

The findings of the current study are limited by the small sample size, and therefore, careful interpretation of the findings of this pilot study is required. Like many previous studies aiming to explore the effectiveness of nutritional support in stable (non-exacerbating) malnourished patients living with COPD, recruitment is a challenge, with meta-analyses showing recruitment in included trials ranging from 9 to 71 participants [11]. Nevertheless, this study was able to recruit enough outpatients (n = 33) to allow pooled analysis for adequate statistical power to investigate changes in outcomes of interest from baseline. The current findings are in line with the existing scientific literature in this area, which consistently demonstrates that malnutrition in COPD is amenable to nutritional support and results in significant clinical improvements and that a treatment duration of 12 weeks does appear adequate to be able to observe measurable improvements. 

### 4.2. Implications for Practice

There are several important implications to practice that arise from this study. The first is that the findings do go some way toward supporting current routine practice for the first line treatment for malnutrition in patients with COPD, which commonly includes ‘food first’ strategies, including individualized dietary counseling by a dietitian. However, in the absence of food and high-energy, high-protein snack provision to outpatients, this study highlights that dietary counseling alone does not result in significant improvements in energy and protein intake from food. What dietary counseling in combination with ONSs (multi-modal intervention) does appear to do is ensure there is no compensatory reduction in energy and protein intake from food, which has been found to be the case in previous studies where dietary counseling or ONSs are used in isolation (reference accompanying Collins Nutrients paper 2024). As a result, two-thirds of the additional energy and protein consumed by patients in the current study came via the powdered ONSs. The current findings are in overall agreement with the most recent Cochrane review looking at dietary advice with or without ONSs in malnourished patients broadly, which suggests dietary counseling plus ONSs to improve weight at 12 weeks (+1.25 kg, 95% CI 0.73 to 1.76 kg) [30]. However, the authors go on to highlight the inconsistencies between studies, preventing firm conclusions from being drawn. Based on the current findings and the literature on nutritional support in treating malnutrition in COPD, multi-modal interventions involving individualized dietary counseling by a dietitian and provision of ONS, either in powdered or ready-to-drink form, would appear to be supported. In the current study, there was a trend for increased reliance on ONSs with COPD severity. This warrants further investigation in future studies, as it might be that patients with more severe respiratory disease benefit from ready-to-eat meals and ready-to-drink ONSs due to physical limitations, whereas for individuals with less severe disease, food fortification and powdered ONSs might be acceptable. Dietitians are uniquely positioned to consider what is likely to be most appropriate at the individual level, incorporate ONSs within nutrition care plans, as indicated, to maximize effectiveness, assist patients with the most cost-effective means of obtaining ONSs within their local context, and adjust as required. This type of approach is seen in recent guidelines on managing malnutrition in COPD [31]. 

Although beyond the scope of the current research, the fact that most outpatients in the current study required a home visit, does limit the scalability of individualized counseling delivered by a dietitian. However, considerable learnings can be taken from the COVID-19 pandemic with regards to nutritional support and respiratory disease [32]. Early, individualized nutrition interventions are important during acute illness with respiratory disease and should be continued across the healthcare continuum (hospital-to-home). Future studies should explore the potential for technology to assist with this transition, improving access to dietitians and ensuring scalable and sustainable services [33,34]. 

## 5. Conclusions

Multimodal nutrition intervention involving powdered ONSs and individualized dietary counseling by a dietitian is effective in treating malnutrition in outpatients with COPD, resulting in improvements in nutritional status and quality of life. Future interventional studies should seek to implement an individualized approach to nutritional support that is facilitated by technology to improve patient-reported experience measures (PREMs) as well as patient-reported outcome measures (PROMs). 

## Figures and Tables

**Figure 1 nutrients-16-01696-f001:**
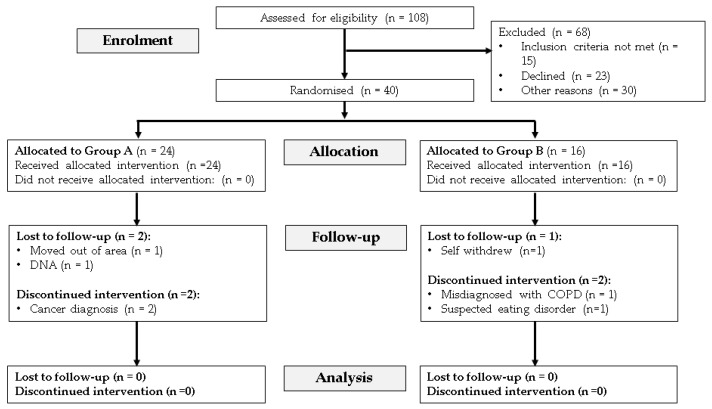
Study flow diagram (Consolidated Standards of Reporting Trials; CONSORT).

**Figure 2 nutrients-16-01696-f002:**
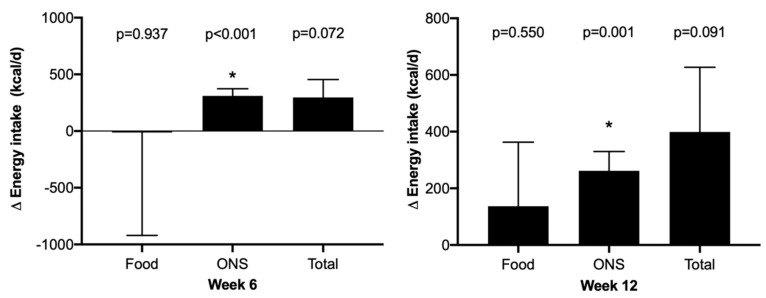
Change in energy intake (kcal/d). Values presented are mean (SE), * = significant change from baseline.

**Figure 3 nutrients-16-01696-f003:**
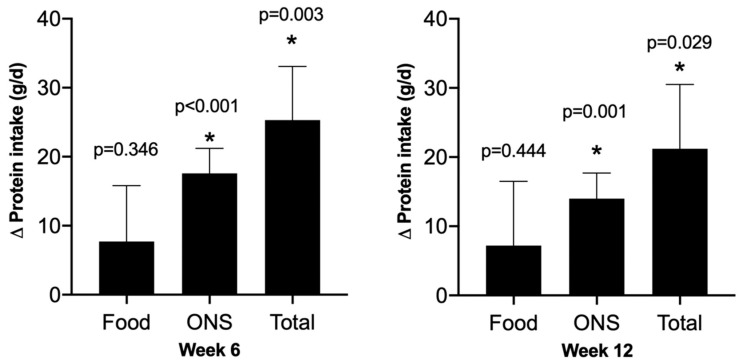
Change in protein intake (g/d). Values presented are mean (SE), * = significant change from baseline. Left panel: week 6 and Right panel: week 12.

**Figure 4 nutrients-16-01696-f004:**
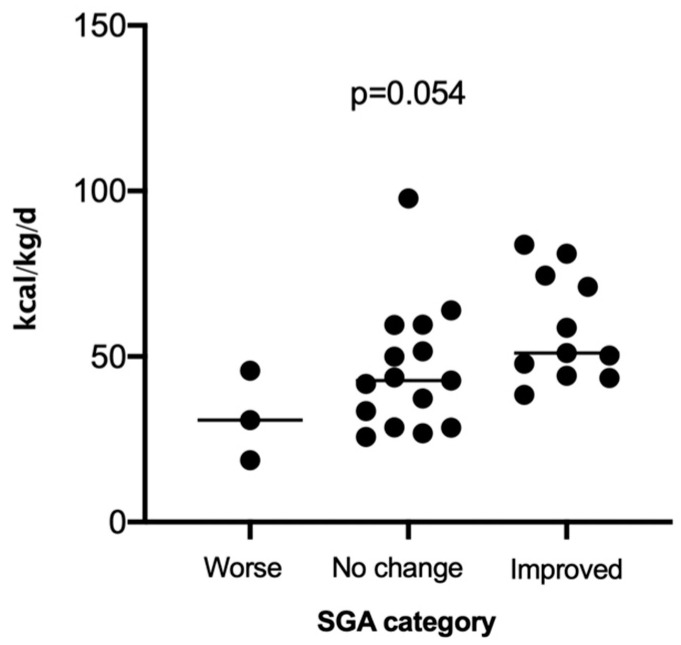
SGA category and energy intake at week 12.

**Table 1 nutrients-16-01696-t001:** Participant characteristics for those completing baseline assessment.

	Group A: DA	Group B: DA + ONS	*p*
**Characteristics**	(*n* 21)	(*n* 12)	
Age (years)	66.6 (12.4)	66.8 (11.1)	0.961
Gender (Male:Female)	11:10	7:5	0.741
Home visit (Yes:No)	15:6	10:2	0.443
**COPD severity (n (%))**			
Moderate	5 (23.8)	0 (0)	
Severe	8 (38.1)	5 (41.7)	
Very severe	8 (38.1)	7 (58.3)	0.168
**Nutritional status & intake**			
Weight (kg)	50.3 (8.5)	51.5 (7.6)	0.690
BMI (kg/m^2^)	17.7 (2.3)	18.7 (2.8)	0.293
Energy intake (kcal/d)	2061 (873)	2171 (704)	0.711
Kcal/kg·BW/d	42.4 (21.6)	42.8 (14.2)	0.963
Protein intake (g/d)	73.0 (32.0)	79.9 (35.3)	0.583
g/kg·BW/d	1.6 (0.8)	1.4 (0.7)	0.498
**Functional capacity**			
Right handgrip (kg)	35.4 (17.3)	33.3 (15.2)	0.733
Left handgrip (kg)	32.8 (14.7)	30.9 (14.8)	0.713
**Quality of life**			
SGRQ score (total)	59.5 (14.1)	57.3 (22.1)	0.737

Group A—DA: dietary advice + written education + recommended to purchase ONS; Group B—ONSs: Dietary advice + written education + ONSs provided; COPD: chronic obstructive pulmonary disease; BW: body weight; SGRQ: St. George’s Respiratory Questionnaire.

## Data Availability

Data available on request due to restrictions. The data presented in this study are available on request from the corresponding author.

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
