# Peer review of "Nutritional Support in Malnourished Outpatients with Chronic Obstructive Pulmonary Disease (COPD): A Randomized Controlled Pilot Study"

_nutrients, 2024, doi:10.3390/nu16111696_

Round 1

Reviewer 1 Report

Comments and Suggestions for Authors

Thank you for the wonderful opportunity to review this manuscript. This is an interesting paper exploring the effectiveness of a multi-modal nutritional intervention commonly used as a first-line treatment for malnutrition in COPD outpatients as well as in other patients’ groups.

The paper is written to a high standard, and I only have few comments.

The results are positive since both groups gained weight, increased intake and there were improvements in quality of life. However, the timing of the intervention (12 weeks) is rather short in a chronic disease such as COPD whereas long time treatment is needed. This needs to be discussed.

The only difference between intervention groups was powdered ONS free of charge for group B. If the study had been adequately powered, I don´t think significant difference between the two groups would have been seen.

Rather than just using body weight as an outcome, body composition measurements, both at baseline and follow-up would have added more to the study.

You used the Subjective Global Assessment to diagnose malnutrition, and while it was not a specific aim of your study, did you thought about using the GLIM diagnostic criteria?

It was nice to offer the follow up home visits which is not always possible but especially important in a disease such as COPD where people are often unable to attend an outpatient clinic.

In total n=108 were assessed for eligibility, n=23 declined and n=30 were excluded due to other reasons. Can you add more information on why some declined and what the other reasons were.

Line 341: Take one „is“ or change one „is“ in the sentence.

Line 410: The reference is yellow.

Author Response

Dear Reviewers, 

Please find attached our point-to-point response. We appreciate the feedback you provided and the opportunity to provide clarification and improve the readability of the manuscript. 

Kind regards.

Reviewer 2 Report

Comments and Suggestions for Authors

The manuscript submitted by Conway et al. investigated the efficacy of powdered oral nutritional supplements (ONS) combined with personalized dietary counselling in managing malnutrition in COPD patients with a randomized trial. If the authors observed no difference between the two studied groups (Dietary advice and Dietary advice + ONS) they showed that adherence to ONS was associated with better outcomes, highlighting the importance of ONS type and delivery method in clinical practice and future research. Overall, nutrition support significantly enhances nutritional status and quality of life in malnourished COPD outpatients.

The data is interesting, but its presentation and analysis are not relevant, which diminishes the scope and interest of the study. Furthermore, inaccuracies and a study design problem are significant weaknesses for this manuscript.

1- Two groups were studied but except for the baseline characteristics no data were given for the follow up and for the groups. These results should be presented in a table. The authors stated that no difference were observed between groups at 6 and 12 weeks for the studied parameters, but comparing the mean values at the endpoint is not the best way to highlight differences. It would have been better to compare for each group the values at baseline and the values at 6-12 weeks.

2- At a statistical level: the analysis aiming at carrying out t-tests is not suitable for this type of study. All patients have longitudinal follow-up, so it is best to compare the values of interest (nutritional status and intake) for each patient before and after treatment and see if there is a difference between the groups in the evolution of these parameters. For this, the appropriate statistical test is an anova two-way test or a mixed model is they are missing values. Such a test allows to pair values before and after treatment in each group and strengthen the analysis.

3- Another major point is the number of patients and their randomization. In methods section 2.4, it is written (lines 143-145): “In order to achieve the statistical power to detect differences in primary endpoints (quality of life and body weight) a target sample size of 100 patients was identified.” However, only 40 patients were randomized in the study. This number is not consistent with the target number and requires explanation.

Additionally, after randomization, the number of subjects is not the same in each group which is normally a randomization criteria. Moreover, the distribution by severity degree of COPD is not homogeneous in the groups which should normally be a important criteria too. We can thus question the effectiveness of the randomization which normally aims to distribute people homogeneously. This heterogeneity seems a major problem, with regard to the parameters studied because it  has been shown that malnutrition in COPD is dependent on the degree of severity (DOI: 10.2147/COPD.S179609; DOI: 10.3390/nu14010044). This difference between groups leads to a bias in the interpretation of the results and this limitation must be discussed.

Usually, COPD severity is classified as defined by Global Initiative for Chronic Obstructive Lung Disease (GOLD). Please use this nomenclature instead of moderate-severe-very severe.

In the same way, is the degree of malnutrition (SGA A and SGA B) identical in each group?

4- How many patients consumed ONS in group A and in group B?

Is weight gain depending on:

a- the severity of COPD (not identical in each group)?

b- the degree of malnutrition? Is it identical in each group?

c- not different between groups because the same number of people consumed ONS?

All These points should be addressed, analyzed and discussed.

5- Another concern is the ‘n’ values throughout the text:

- in figure 1, 40 patients were randomized (24 in group A and 16 in group B) with 4 eliminations in group A (n=20) and 3 in group B (n=13). In table 1 n values are 21 in group A and 12 in group B. Please correct or explain.

- line 186: ‘Analysis of the whole cohort (Group A + Group B (n 24), paired t-test) revealed nutrition 186 support was associated…’ Please explain to what n=24 refers because the whole cohort is n=33. N=24 seemed to correspond to malnourished and severely malnourished people. Did the autors referred to that population?

Author Response

Dear Reviewers, 

Please find attached the point-to-point response to the feedback received. We thank you for your time in thoroughly reviewing the manuscript and the opportunity to clarify the points raised and improve the readability of the manuscript. 

Kind regards. 

Round 2

Reviewer 1 Report

Comments and Suggestions for Authors

I have no further comments for authors and accept the manuscript in present form.